# EXPLAINABILITY FOR FAIR MACHINE LEARNING

## ABSTRACT

As the decisions made or influenced by machine learning models increasingly impact our lives, it is crucial to detect, understand, and mitigate unfairness. But even simply determining what "unfairness" should mean in a given context is non-trivial: there are many competing definitions, and choosing between them often requires a deep understanding of the underlying task. It is thus tempting to use model explainability to gain insights into model fairness, however existing explainability tools do not reliably indicate whether a model is indeed fair. In this work we present a new approach to explaining fairness in machine learning, based on the Shapley value paradigm. Our fairness explanations attribute a model's overall unfairness to individual input features, even in cases where the model does not operate on sensitive attributes directly. Moreover, motivated by the linearity of Shapley explainability, we propose a meta algorithm for applying existing training-time fairness interventions, wherein one trains a perturbation to the original model, rather than a new model entirely. By explaining the original model, the perturbation, and the fair-corrected model, we gain insight into the accuracy-fairness trade-off that is being made by the intervention. We further show that this meta algorithm enjoys both flexibility and stability benefits with no loss in performance.

## 1 INTRODUCTION

Machine learning has repeatedly demonstrated astonishing predictive power due to its capacity to learn complex relationships from data. However, it is well known that machine learning models risk perpetuating or even exacerbating unfair biases learnt from historical data (Barocas & Selbst, 2016; Bolukbasi et al., 2016; Caliskan et al., 2017; Lum & Isaac, 2016). As such models are increasingly used for decisions that impact our lives, we are compelled to ensure those decisions are made fairly.

In the pursuit of training a fair model, one encounters the immediate challenge of how fairness should be defined. There exist a wide variety of definitions of fairness — some based on statistical measures, others on causal reasoning, some imposing constraints on group outcomes, others at the individual level — and each notion is often incompatible with its alternatives (Berk et al., 2018; Corbett-Davies et al., 2017; Kleinberg et al., 2017; Lipton et al., 2018; Pleiss et al., 2017). Deciding which measure of fairness to impose thus requires extensive contextual understanding and domain knowledge. Further still, one should understand the downstream consequences of a fairness intervention before imposing it on the model's decisions (Hu et al., 2019; Liu et al., 2018).

To help understand whether a model is making fair decisions, and choose an appropriate notion of fairness, one might be tempted to turn to model explainability techniques. Unfortunately, it has been shown that many standard explanation methods can be manipulated to suppress the reported importance of the protected attribute without substantially changing the output of the model (Dimanov et al., 2020). Consequently such explanations are poorly suited to assessing or quantifying unfairness.

In this work, we introduce new explainability methods for fairness based on the Shapley value framework for model explainability (Datta et al., 2016; Štrumbelj & Kononenko, 2010; Lipovetsky & Conklin, 2001; Lundberg & Lee, 2017; Štrumbelj & Kononenko, 2014). We consider a broad set of widely applied group-fairness criteria and propose a unified approach to explaining unfairness within any one of them. This set of fairness criteria includes *demographic parity*, *equalised odds*, *equal opportunity* and *conditional demographic parity* see Sec. 2.1. We show that for each of these definitions it is possible to choose Shapley value functions which capture the overall unfairness in the model, and attribute it to individual features. We also show that because the fairness Shapley values collectively must sum to the chosen fairness metric, we cannot hide unfairness by manipulating the explanations of individual features, thereby overcoming the problems with accuracy-based explanations observed by Dimanov et al. (2020).

Motivated by the attractive linearity properties of Shapley value explanations, we also introduce a meta algorithm for training a fair model. Rather than learning a fair model directly, we propose instead learning an additive correction to an existing unfair model. We use training-time fairness algorithms to train the correction, thereby ensuring the corrected model is fair. We show that this approach gives new perspectives helpful for understanding fairness, benefits from greater flexibility due to model-agnosticism, and enjoys improved stability, all while maintaining the performance of the chosen training-time algorithm.

## 2  EXPLAINABLE FAIRNESS

In this section we give an overview of the Shapley value paradigm for machine learning explainability, and show how it can be adapted to explain fairness. Motivated by the axiomatic properties of Shapley values, we also introduce a meta algorithm for applying training-time fairness algorithms to a perturbation rather than a fresh model, giving us multiple perspectives on fairness.

### 2.1  BACKGROUND AND NOTATION

We consider fairness in the context of supervised classification, where the data consists of triples $(x, a, y)$, where $x \in \mathcal{X}$ are the features, $a \in \mathcal{A}$ is a protected attribute (e.g. sex or race), and $y \in \mathcal{Y}$ is the target. We allow, but do not require, $a$ to be a component of $x$. The task is to train a model $f$ to predict $y$ from $x$ while avoiding unfair discrimination with respect to $a$. We assume $\mathcal{A}$ and $\mathcal{Y}$ are both finite, discrete sets.

Our fairness explanations apply to any definition that can be formulated as (conditional) independence of the model output and the protected attribute. This includes *demographic parity* (Calders et al., 2009; Feldman et al., 2015; Kamiran & Calders, 2012; Zafar et al., 2017), *conditional demographic parity* (Corbett-Davies et al., 2017), and *equalised odds* and *equal opportunity* (Hardt et al., 2016).

**Definition 1.** DEMOGRAPHIC PARITY *The model $f$ satisfies demographic parity if $f(x)$ is independent of $a$, or equivalently $P(f(x) = \tilde{y}|a) = P(f(x) = \tilde{y})$ for all $\tilde{y} \in \mathcal{Y}$ and $a \in \mathcal{A}$.*

**Definition 2.** CONDITIONAL DEMOGRAPHIC PARITY *The model $f$ satisfies conditional demographic parity if with respect to a set of* legitimate risk factors $\{v_1, \ldots, v_n\}$ *if $f(x)$ is independent of $a$ conditional on the $v_i$, or equivalently $P(f(x) = \tilde{y}|a, v_1, \ldots, v_n) = P(f(x) = \tilde{y}|v_1, \ldots, v_n)$ for all $\tilde{y} \in \mathcal{Y}$ and $a \in \mathcal{A}$.*

**Definition 3.** EQUALISED ODDS *The model $f$ satisfies equalised odds if $f(x)$ is independent of $a$ conditional on $y$, or equivalently $P(f(x) = \tilde{y}|a, y) = P(f(x) = \tilde{y}|y)$ for all $\tilde{y}, y \in \mathcal{Y}$ and $a \in \mathcal{A}$.*

If $\mathcal{Y} = \{0, 1\}$ is binary, then equalised odds implies that the true and false positive rates on each protected group should agree. Furthermore, assuming that $y = 1$ corresponds to the "privelidged outcome", we can define equal opportunity as follows

**Definition 4.** EQUAL OPPORTUNITY *The model $f$ satisfies equal opportunity if $f(x)$ is independent of $a$ conditional on $y = 1$, or equivalently $P(f(x) = \tilde{y}|a, y = 1) = P(f(x) = \tilde{y}|y = 1)$ for all $\tilde{y} \in \mathcal{Y}$ and $a \in \mathcal{A}$.*

### 2.2  ADAPTING EXPLAINABILTY TO FAIRNESS

Fairness in decision making – automated or not – is a subtle topic. Choosing an appropriate definition of fairness requires both context and domain knowledge. In seeking to improve our understanding of the problem, we might be tempted to use model explainability methods. However Dimanov et al. (2020) show that such methods are poorly suited for understanding fairness. In particular we should not try to quantify unfairness by looking at the feature importance of the protected attribute, as such measures can be easily manipulated. Part of the problem is that most explainability methods attempt to determine which features are important contibutors to the model's accuracy. We seek to introduce explanations that instead determine which features contributed to unfairness in the model.

Toward this end, we work within the Shapley value paradigm, which is widely used as a model-agnostic and theoretically principled approach to model explainability (Datta et al., 2016; Štrumbelj & Kononenko, 2010; Lipovetsky & Conklin, 2001; Lundberg & Lee, 2017; Štrumbelj & Kononenko, 2014). We will first review the application of Shapley values to explaining model accuracy, then show how this can be adapted to explaining model unfairness. See Frye et al. (2020b) for a detailed analysis of the axiomatic foundations of Shapley values in the context of model explainability.

Shapley values provide a method from cooperative game theory to attribute value to the individual players on a team $N = \{1, \ldots, n\}$ (Shapley, 1953). If the team earns a total value $v(N)$, the Shapley value $\phi_v(i)$ attributes a portion to player $i$ according to:

$$\phi_v(i) = \sum_{S \subseteq N \setminus \{i\}} \frac{|S|!\,(n - |S| - 1)!}{n!} \big[v(S \cup \{i\}) - v(S)\big] \tag{1}$$

Here $v(S)$ is the value a coalition $S$ of players generates when playing on their own. The Shapley value $\phi_v(i)$ is thus the average marginal contribution that player $i$ makes upon joining a coalition, averaged over all coalitions and all orders in which those coalitions can form.

To apply Shapley values to model explainability, one interprets the input features as the players of the game and defines an appropriate value function (e.g. the model's output) to insert into Eq. (1). Let $f_y(x)$ denote the predicted probability that $x$ belongs to class $y$. We define a value function by marginalising over out-of-coalition features:

$$v_{f_y(x)}(S) = \mathbb{E}_{p(x')}\big[f_y(x_S \sqcup x'_{N \setminus S})\big] \tag{2}$$

where $x_S$ is the set of feature values with indices in $S$, $x_S \sqcup x'_{N \setminus S}$ is a new data point formed by filling the missing features in $x_S$ with values from $x'$, and where $p(x')$ represents the data distribution.[1] One computes local Shapley values $\phi_{f_y(x)}(i)$ by inserting $v_{f_y(x)}$ into Eq. (1). These can be aggregated to obtain a global explanation of the model that maintains the underlying Shapley axioms:

$$\Phi_f(i) = \mathbb{E}_{p(x,y)}\big[\phi_{f_y(x)}(i)\big] \tag{3}$$

where $p(x, y)$ is the joint distribution from which the labelled data is sampled, and so $f_y(x)$ is the probability the model assigns to the true outcome. Aggregating global Shapley values in this way provides the desirable property that

$$\sum_i \Phi_f(i) = \mathbb{E}_{p(x,y)}\big[f_y(x)\big] - \mathbb{E}_{p(x')p(y)}\big[f_y(x')\big] \tag{4}$$

The first term on the right-hand side is the average probability assigned to the true outcome. It can be interpreted as the expected accuracy of a randomised classifier that samples a predicted label according to the probabilities predicted by the model. The second is an offset term corresponding to the expected accuracy if we were to sample a predicted label at random according to the average prediction probabilities for each class. This offset is not attributable to any of the features and is related to the class balance. We remark that randomised classifiers are often used when training fair models, for example by the reductions approach of Agarwal et al. (2018), so expected accuracy coincides with commonly used deterministic accuracy. More generally, the expected accuracy is closely related to usual notions of accuracy, but additionally captures the confidence with which the classifier makes predictions.

EXPLAINING MODEL FAIRNESS

To explain fairness in a model's decisions, we define a new value function that captures fairness, then proceed as in Sec. 2.2. To simplify the exposition, we assume that $\mathcal{Y} = \mathcal{A} = \{0, 1\}$. See App. A.2 for straightforward generalisations to other notions of fairness, multiclass classification, and non-binary protected attribute. Without loss of generality we assume $f(x)$ is scalar and corresponds to the predicted probability that $y = 1$. To begin, let

$$g_a(x) = f(x) \cdot \frac{(-1)^a}{p(a)} \tag{5}$$

where $p(a)$ denotes the proportion of individuals with protected attribute $a$, and in which the sign of $g_a(x)$ is controlled by the true value of $a$, whether or not the $a$-component in $x$ is altered. The value function on coalitions is defined through marginalisation:[2]

$$v_{g_a(x)}(S) = \mathbb{E}_{p(x')}\big[g_a(x_S \sqcup x'_{N \setminus S})\big] \tag{6}$$

---

[1] A more natural value function would marginalise over the conditional data distribution $p(x'_{N \setminus S}|x_S)$ as discussed in Aas et al. (2019) and Frye et al. (2020a). Eq. (2) is used here for clarity of exposition as it is more commonly found in the literature.

[2] This marginalisation could also be performed conditionally, as discussed in footnote 1.

Next, we insert $v_{g_a(x)}$ into Eq. (1) to obtain local Shapley values $\phi_{g_a(x)}(i)$. Finally, we obtain the corresponding global Shapley values through aggregation:

$$\Phi_g(i) = \mathbb{E}_{p(x,a)}\big[\phi_{g_a(x)}(i)\big] \tag{7}$$

where $p(x,a)$ is the joint distribution of features and protected attribute from which the data is sampled. The definitions above are motivated by the resulting sum rule for the global Shapley values:

$$\sum_i \Phi_g(i) = \int dx\, p(x|a=0)\, f(x) - \int dx\, p(x|a=1)\, f(x) \tag{8}$$

This is the expected demographic parity difference for randomised classifier that samples labels according to the model's predicted probabilities. The global Shapley values $\Phi_g(i)$ can thus be interpreted as each feature's marginal contribution to the overall demographic parity difference. The local Shapley values are less relevant, as group notions of fairness are defined in terms of aggregate statistics over each group, and local Shapley values only explain individual predictions.

## 2.3 LEARNING CORRECTIVE PERTURBATIONS

The Shapley linearity axiom guarantees that the fairness Shapley values of a sum of models is the sum of the fairness Shapley values for each model. Motivated by this we consider the problem of learning an additive perturbation to an existing model in order to impose fairness. That is, given $f$ we would like to learn a parametric perturbation $\delta_\theta$ such that

$$f_\theta(x) = f(x) + \delta_\theta(f(x), a, x) \tag{9}$$

is fair. We would then gain three different perspectives from the explanations of each component: the original sources of unfairness from $f$, the correction or trade-off that was made from $\delta_\theta$, and how the corrected model makes fair predictions from $f_\theta$. Moreover, the explanations are mutually consistent.

We continue to focus on binary classification here, see App. A.1 for a generalisation to the multiclass classification case. Since we require that the output of the corrected model should also be a valid probability, we define $\delta_\theta$ via an auxiliary perturbation $\tilde{\delta}_\theta$ on the logit scale:

$$\delta_\theta(f(x), x, a) = \sigma\Big(\sigma^{-1}\big(f(x)\big) + \tilde{\delta}_\theta\big(f(x), x, a\big)\Big) - f(x). \tag{10}$$

where $\sigma(x) = (1 + \exp(-x))^{-1}$. We can apply a training-time fairness algorithm of our choice to $f_\theta$ in order to learn the parameters of the auxiliary perturbation $\tilde{\delta}_\theta$. In Sec. 3 we show results for the algorithms of Agarwal et al. (2018) and Zhang et al. (2018) applied to the perturbed model.

Note that while in Eq. (10) we write $\delta_\theta$ as a function of $x$, $a$ and $f(x)$, it is possible to use only a subset of these inputs. If we only gave $\delta_\theta$ access to $a$ and $f(x)$ then this becomes a true post-processing algorithm. Alternatively we may require a fair algorithm that does not have access to the protected attribute at inference time, in which case we could omit $a$. We would then need to select a training-time algorithm that is able to make predictions without access to the protected attribute.

## 3 RESULTS

In this section we showcase the fairness Shapley values introduced in Sec. 2. We further show that the models corrected through learning a perturbation suffer no loss in performance when compared to their unconstrained analogues, and enjoy increased flexibility and stability in addition to the explainability benefits. Our experiments are conducted on the Adult dataset from the UCI Machine Learning Repository (Dua & Graff, 2017), where the task is to predict whether an individual earns more than $50K per year based on their demographics, and the COMPAS recidivism dataset (Larson et al., 2016), where the task is to predict recidivism risk based on demographics. We use sex as the protected attribute in our experiments on Adult, and race in our experiments on COMPAS. Full details for all experiments are in App. A.3.

## 3.1 EXPLAINABILITY

To test the effectiveness of the fairness Shapley values, we train a baseline-unfair model on the Adult dataset, then train a correction by applying the adversarial approach of Zhang et al. (2018) to an additive perturbation as described in Sec. 2. From left to right Fig. 1 shows the accuracy and fairness

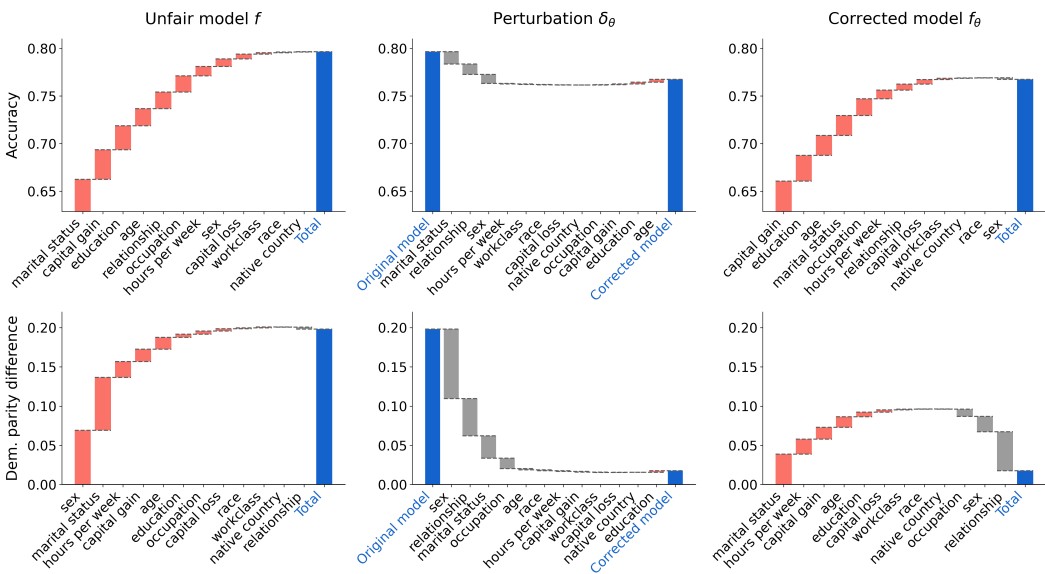

Figure 1: Explaining accuracy and unfairness (demographic parity) using Shapley values.

global Shapley values for the original model, perturbation, and corrected model respectively. We use waterfall plots to emphasise that they sum to the expected model accuracy or expected demographic parity difference respectively. The offset term for the accuracy Shapley values described in Sec. 2 is used to offset the $y$-axis in the top row, fairness Shapley values have no offset term.

The accuracy Shapley values tell us which features contribute to the model accuracy, while the fairness Shapley values tell us which features contribute to the demographic parity difference. Comparing the two can give us useful insights into the legitimacy of a feature. For example we see that `marital-status` is the largest individual contributor to accuracy for the unfair model, but also makes a major contribution to unfairness. Weighing up the effects of each feature on both accuracy and fairness helps us determine whether it is legitimate to include them in the model.

We gain different insights into the problem from each of the three models. The explanation of the original-unfair model shows us why an unconstrained model might learn to make predictions in an unfair way, the explanation of the corrected-fair model shows us how it is able to make fair predictions, and the explanation of the perturbation captures the trade-off that has been made to impose fairness.

For example, in Fig. 1 we see that the unfair model uses all features to improve accuracy, as we would expect from a model that has been trained only to maximise accuracy. The accuracy Shapley values of the perturbation show that some predictive power from `sex` and its proxies `marital-status` and `relationship` has been sacrificed, but the fairness Shapley values show that as a result these three features were able to be used to heavily reduce the demographic parity difference. The fairness Shapley values for the corrected model show that the effects of `marital-status` and other features are almost perfectly counter-balanced by `sex` and `relationship`.

The complementary fairness and accuracy explanations, and the explanations offered from the three models have the potential to be a powerful tool for model development and fine-tuning of fairness. In App. A.5 we repeat this experiment with a model that does not have access to the protected attribute and compare results.

## 3.2  ROBUSTNESS OF FAIRNESS EXPLANATIONS

Dimanov et al. (2020) show that existing explainability methods are poorly suited to answering questions about fairness. Specifically they show that by retraining a model with an additional penalty term corresponding to the influence the protected attribute has on the output, the explanation for that feature can be suppressed without substantially affecting the model predictions. Hence the feature importance of the protected attribute is a poor measure of fairness.

Fig. 2 shows the accuracy and fairness Shapley values for the baseline model of the previous section, as well as the same model after retraining to suppress the influence of `sex` on the output. We see

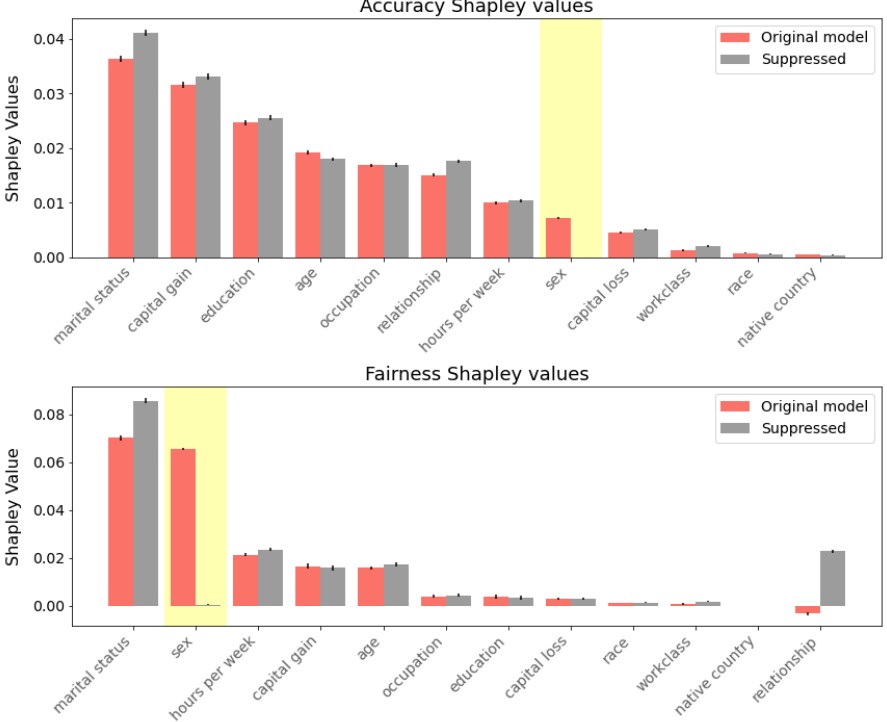

Figure 2: Suppressing the importance of the `sex` feature.

that retraining has completely eliminated *functional* dependence of the output on `sex`, however we should not conclude that the model now satisfies demographic parity. Indeed the two models predict the same outcome on more than 98% of the test data points, and demographic parity difference only improved fractionally, falling from 0.193 to 0.184.

The fairness Shapley values give us a more complete picture. Because the fairness Shapley values must sum to the expected demographic parity difference, they will always capture unfairness if present. If we suppress unfairness attributed to any one feature, the attributions of other features must increase to compensate. The only way to hide unfairness from the fairness Shapley values is to eliminate it.

Since we have manipulated $f$ to no longer depend on `sex`, the fairness Shapley values for `sex` has also been suppressed. However, the fairness Shapley values for `marital-status` and `relationship` have increased, showing that the model has simply shifted focus to proxies of the protected attribute. We see the same effect in the accuracy Shapley values, however without the context of its impact on fairness it's hard to know whether the increased accuracy attributable to `relationship` is legitimate.

## 3.3 LEARNT PERTURBATIONS

In this section we investigate the properties of applying existing training-time algorithms to the additive perturbations of Sec. 2.3. We find that applying the reductions approach of Agarwal et al. (2018) and the adversarial approach of Zhang et al. (2018) to a perturbation leads to no loss in performance compared to applying the same algorithms to a new model, but we gain flexibility, stability and the benefit of being able to compare explanations between original model, perturbation and corrected model.

### PERFORMANCE

We proposed the learnt perturbations in Sec. 2 primarily due to their explainability benefits. It is important however that we verify that in doing so we have not compromised the performance of the algorithms. In order check this we use the adversarial approach of Zhang et al. (2018) and the reductions approach of Agarwal et al. (2018) to impose both demographic parity and equalised odds

Table 1: Accuracy associated with decreasing demographic parity thresholds.

| | Method | Accuracy [%] at demographic parity difference | | | | | | |
|---|---|---|---|---|---|---|---|---|
| | | 0.1 | 0.08 | 0.06 | 0.04 | 0.02 | 0.01 | 0.005 |
| **Adult** | Agarwal et al. | 84.71 | 84.32 | 83.94 | 83.82 | 83.29 | 83.29 | - |
| | Agarwal et al. - perturbed | 84.69 | 84.43 | 83.82 | 83.82 | 83.35 | 83.23 | - |
| | Zhang et al. | 84.65 | 84.18 | 84.06 | 83.58 | 83.18 | 83.15 | 83.15 |
| | Zhang et al. - perturbed | 84.74 | 84.48 | 83.78 | 83.61 | 83.14 | 82.99 | 82.96 |
| | Feldman et al. (post) | 84.69 | 84.35 | 84.12 | 83.67 | 83.32 | 83.30 | 83.01 |
| **COMPAS** | Agarwal et al. | 74.05 | 74.05 | 73.77 | 73.67 | 73.11 | 73.11 | 73.01 |
| | Agarwal et al. - perturbed | 74.24 | 74.24 | 73.86 | 73.86 | 73.20 | 72.73 | 72.73 |
| | Zhang et al. | 75.19 | 75.19 | 75.19 | 74.62 | 74.15 | 74.15 | 74.15 |
| | Zhang et al. - perturbed | 74.24 | 74.24 | 74.24 | 73.30 | 73.30 | 73.20 | 72.73 |
| | Feldman et al. (post) | 74.81 | 74.81 | 74.81 | 74.24 | 74.24 | 73.20 | 72.35 |

Table 2: Accuracy associated with decreasing equalised odds thresholds.

| | Method | Accuracy [%] at equalised odds difference | | | | | | |
|---|---|---|---|---|---|---|---|---|
| | | 0.1 | 0.08 | 0.06 | 0.04 | 0.02 | 0.01 | 0.005 |
| **Adult** | Agarwal et al. | 85.32 | 85.32 | 85.13 | 84.30 | 84.18 | - | - |
| | Agarwal et al. - perturbed | 85.43 | 85.43 | 85.31 | 84.34 | 84.21 | - | - |
| | Zhang et al. | 85.13 | 85.04 | 85.04 | 84.86 | 84.33 | 75.43 | 75.43 |
| | Zhang et al. - perturbed | 85.26 | 85.11 | 85.06 | 84.97 | 84.23 | 83.53 | - |
| | Hardt et al. | 82.77 | 82.77 | 82.77 | 82.77 | 82.77 | 82.77 | 82.77 |
| **COMPAS** | Agarwal et al. | 75.19 | 75.19 | 74.05 | 74.05 | 73.86 | 73.39 | 73.39 |
| | Agarwal et al. - perturbed | 74.43 | 74.43 | 74.43 | 73.86 | 73.39 | 73.39 | 73.39 |
| | Zhang et al. | 74.62 | 74.62 | 74.62 | 74.62 | 74.62 | 73.48 | 53.12 |
| | Zhang et al. - perturbed | 74.34 | 74.34 | 74.34 | 74.34 | 73.48 | 72.44 | 72.44 |
| | Hardt et al. | 71.31 | 71.31 | 71.31 | 70.45 | 68.75 | - | - |

on the Adult and COMPAS datasets. In both cases we apply the algorithms to new models, and also to perturbed models as described in Sec. 2.3. We also compare to simpler post-processing approaches of Feldman et al. (2015) for demographic parity, and Hardt et al. (2016) for equalised odds.

Table 1 shows the results of our demographic parity experiments, and Table 2 shows the results of the equalised odds experiments. We report for each method the highest observed test-set accuracy where the corresponding fairness metric did not exceed a specified threshold. All methods are seen to lose accuracy as the fairness requirement becomes more stringent, some methods performing better on one task than the other. The perturbed models perform competitively on both datasets and for both definitions of fairness, showing no significant reduction in accuracy at each fairness threshold over their training-time counterparts. For demographic parity we note that while the simple post-processing method of Feldman et al. (2015) performs competitively with the more sophisticated algorithms, it requires access to the protected attribute at inference time, which is not true of the approaches of Agarwal et al. (2018) and Zhang et al. (2018) which only require access to the protected attribute during training.

## FLEXIBILITY

The meta algorithm presented in Sec. 2 offers flexibility benefits that are typical of post-processing algorithms while inheriting the performance benefits of the training-time algorithms it utilises to impose fairness. Specifically, we can be fully model-agnostic with respect to the original model, as any model structure or access requirements of the training-time algorithm apply only to the perturbation, and not the original model. Indeed we could learn a perturbation to a model that is only available for inference over a network, or correct a rules based classifier that does not lend itself to conventional optimisation techniques. Furthermore, if the original model is complex, we have the option of training a lightweight perturbation to the complex model, and may not need to rerun an expensive training procedure.

## STABILITY

Since accuracy and fairness objectives are typically in tension, some optimisation methods exhibit training-time instability where the classifier does not converge to a local minimum, but instead jumps

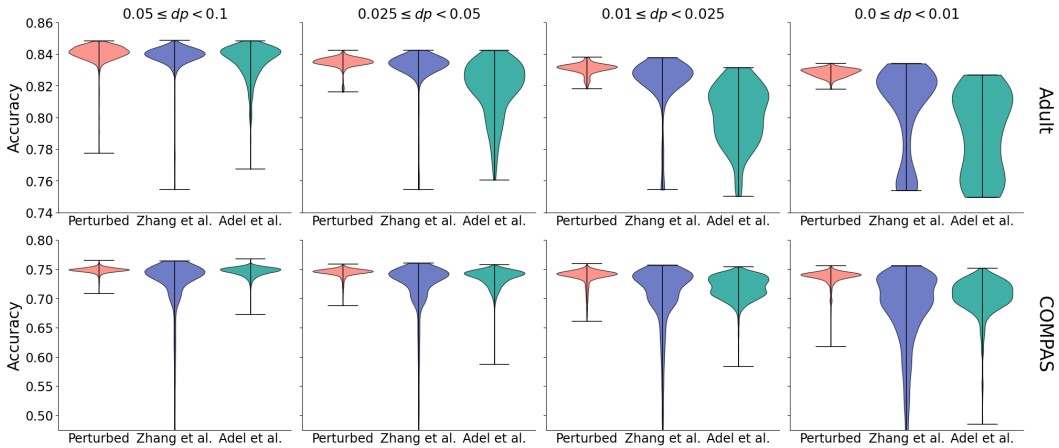

Figure 3: Accuracy violin plots of experimental outcomes binned by achieved level of fairness.

between them. This is particularly true of the adversarial method of Zhang et al. (2018), where the tension between accuracy and fairness manifests itself as the tension between the model and the adversary. Indeed training-time stability is a generic problem for adversarial methods. Adel et al. (2019) introduce a variant of adversarial debiasing that attempts to address this instability.

Our perturbation-based approach combats the instability introduced by this tension between competing objectives: the fair-corrected model remains anchored near the original-unfair model, which has already found a local-accuracy optimum. This is also evident in Table 2 where we observe that as we make the fairness threshold more stringent, classifiers trained with the method of Zhang et al. (2018) collapse to a constant predictor, predicting according to class balance, whereas the perturbed model, optimised in the same way, does not exhibit the same behaviour and is able to learn a non-trivial classifier with low equalised odds difference.

We illustrate the improved stability by comparing the application of adversarial training to a perturbation, and the original approaches of Zhang et al. (2018) and Adel et al. (2019). For each approach we train multiple models with a variety of hyperparameter choices. The resulting outcomes show the spread of model behaviour during training. We visualise the results in Fig. 3, which shows the distribution of accuracy outcomes in every experiment, binned by the achieved level of fairness. The plots show that while the optimal accuracy in each bin is similar across the three methods, our perturbative approach generally has less variance and higher mean accuracy across the range of hyperparameters considered. We found similar results when imposing equalised odds (see App. A.4) thus showing a clear training-time stability advantage of our perturbation-based approach across multiple notions of fairness.

## 4  CONCLUSIONS

In this work we introduced a new approach to explaining fairness in machine learning, based on the Shapley value paradigm. The fairness Shapley values are able to attribute unfairness in a model's predictions to individual features, and complement the already well-used accuracy Shapley values. We also showed that because these explanations directly capture unfairness through their sum, they are robust to manipulation. Additionally, motivated by the linearity properties enjoyed by Shapley values, we introduced a meta algorithm for learning a fairness-imposing perturbation to an unfair model using existing training-time algorithms for fairness. This gives us new perspectives from the corresponding explanations: we are able to explain the unfair model to understand why model predictions might be at risk of being unfair, the corrected model to understand how it is able to make fair predictions, and the perturbation to understand the trade-off that has been made between the two models. Moreover, we showed that training a perturbation of this kind results in no loss in performance compared to training a fresh model, but increases flexibility and can improve the training-time stability by anchoring the corrected model to a local-accuracy optimum. We believe that our approach to explaining fairness will be highly valuable to practitioners, and we hope it will contribute to the development of fairer models, and a greater understanding of unfairness in different machine learning problems.

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

## A  Appendix

### A.1  Perturbing a multiclass classifier

In Sec. 2 we introduced a meta algorithm for correcting an unfair model by learning a perturbation assuming a binary classification task, and remarked that it was straight-forward to generalise to multiclass classification. In this section we give the details.

We assume labels $y \in \{1, \ldots, k\}$ and a model $f : \mathcal{X} \to [0,1]^k$ that maps each datapoint $x$ to a categorical probability distribution over the $k$ classes. We denote by $f_y(x)$ the probability that $x$ belongs to class $y$. By assumption, we have $\sum_{y=1}^k f_y(x) = 1$ for all $x$.

As in Eq. (9), we seek to introduce a perturbation $\delta_\theta$ to produce a fair model $f_\theta$. As in Eq. (10) we will define $\delta_\theta$ via an auxiliary perturbation on an unconstrained scale in order that we maintain valid probabilities. It is conventional to use the softmax function to convert a $k$-vector $z$ of logits to a categorical probability distribution

$$\mathrm{softmax}(z)_i = \frac{\exp(z_i)}{\sum_j \exp(z_j)} \tag{11}$$

The softmax function is invariant under addition of a constant to all arguments. As a result, there is a translation invariance in the space of perturbations, which is not desirable for efficient optimisation. We circumvent this issue by pinning the first logit to zero. Specifically we define

$$l(s)_i := \log(s_i) - \log(s_1) \quad i = 1, \ldots, k \tag{12}$$

which as one easily verifies is a right-inverse for $\mathrm{softmax}$ on the space of $k$-simplices, i.e. $\mathrm{softmax}(l(s)) = s$ for any $k$-simplex $s$. With this in hand we define the perturbation analogously to Eq. (10) as follows

$$\delta_\theta(f(x), x, a) = \mathrm{softmax}(l(f(x)) + \tilde{\delta}_\theta(f(x), x, a)) - f(x) \tag{13}$$

where we require that $(\tilde{\delta}_\theta)_1 \equiv 0$. Thus we perturb only $k - 1$ of the logits for a $k$-category classification problem. With the corrected model defined, we proceed precisely as before, applying a training-time algorithm to $f_\theta$ in order to learn the parameters of $\delta_\theta$.

We note that this setup is easily seen to be equivalent to the formulation in Eq. (10) in terms of the sigmoid and logit functions when $k = 2$. Indeed in that case we have $l(z)_1 = 0$ for all $z$, and $l(z)_2 = \log(z_2) - \log(z_1) = \log(z_2) - \log(1 - z_2) = \sigma^{-1}(z_2)$. Moreover by choice of $\tilde{\delta}_\theta$ the first component of the argument of $\mathrm{softmax}$ on the right hand side of Eq. (13) is 0, and we have $\mathrm{softmax}((0, z))_2 = e^z(1 + e^z)^{-1} = \sigma(z)$.

### A.2  Explainability for other fairness notions

To generalise the fairness explanations to other notions of fairness, we need only to adjust the weighting in Eq. (5) and the distribution in Eq. (7) over which we aggregate to produce global Shapley values. We present the details below. As in App. A.1 we assume a multiclass classification task where $f_y(x)$ denotes the probability that $x$ belongs to class $y$.

#### Equalised odds

The fairness Shapley values presented in Sec. 2 can also be used to determine each feature's marginal contribution to equality of odds. Specifically we modify Eq. (5) to create new functions

$$g_{a,y}(x) = \begin{cases} \frac{f_y(x)}{P(a=0|y)} & \text{if } a = 0 \\ -\frac{f_y(x)}{P(a\neq0|y)} & \text{if } a \neq 0 \end{cases} \tag{14}$$

for each $y$. As before we then create a value function by marginalising over out-of-coalition features

$$v_{g_{a,y}(x)}(S) = \mathbb{E}_{p(x')} \left[ g_{a,y}(x_S \sqcup x'_{\bar{S}}) \right]. \tag{15}$$

Shapley values are then calculated using Eq. (1) to average over all possible coalitions. To compute global Shapley values we aggregate using Eq. (7), but replacing $p(x, a)$ with $p(x, a|y)$, from which we obtain the following analogue of Eq. (8)

$$\sum_i \Phi_{g_y}(i) = \int dx \, p(x|y, a = 0) \, f_y(x) - \int dx \, p(x|y, a \neq 0) \, f_y(x) \tag{16}$$

This corresponds to the (signed) difference in expected sensitivity to label $y$ between the protected group $a = 0$ and all other groups for a classifier that samples predicted labels according to the predicted probabilities. In the case of binary classification, we can interpret each integral as the expected true positive and true negative rates when $y = 1$ and $y = 0$ respectively.

For equality of odds, we proceed exactly as above, but we are concerned only with the case where $y = 1$.

CONDITIONAL DEMOGRAPHIC PARITY

We proceed similarly to obtain explanations of fairness for conditional demographic parity by first defining

$$g_{a,v_1,\ldots,v_n}(x) = \begin{cases} \frac{f_y(x)}{P(a=0|v_1,\ldots,v_n)} & \text{if } a = 0 \\ -\frac{f_y(x)}{P(a\neq 0|v_1,\ldots,v_n)} & \text{if } a \neq 0 \end{cases} \tag{17}$$

for each possible combination of resolving variables $v_1, \ldots, v_n$ and value of $y$, then creating a value function by marginalising over out-of-coalition features

$$v_{g_{a,y,v_1,\ldots,v_n}(x)}(S) = \mathbb{E}_{p(x')} \left[ g_{a,y,v_1,\ldots,v_n}(x_S \sqcup x'_{\bar{S}}) \right]. \tag{18}$$

Shapley values are then calculated using Eq. (1) to average over all possible coalitions. To compute global Shapley values we aggregate using Eq. (7), but replacing $p(x, a)$ with $p(x, a|v_1, \ldots, v_n)$, from which we obtain the following analogue of Eq. (8)

$$\sum_i \Phi_{g_{y,v_1,\ldots,v_n}}(i) = \int dx\, p(x|a = 0, v_1, \ldots, v_n)\, f_y(x) - \int dx\, p(x|a \neq 0, v_1, \ldots, v_n)\, f_y(x) \tag{19}$$

which represents the signed expected conditional demographic parity difference for the protected groups $a = 0$ and $a \neq 0$ (conditional on $v_1, \ldots, v_n$).

A.3   EXPERIMENTAL SETUP

In this section we give details on the datasets and experimental results presented in Sec. 3. All experiments were written in Python, we used TensorFlow to specify and train the models (Abadi et al., 2016) and ran all of our experiments on AWS EC2 instances.

DATASETS

**Adult**

The Adult dataset (Dua & Graff, 2017) contains 48842 rows with 14 features, and is pre-split into 66.7% training and 33.3% testing data. We drop the `fnlwgt` feature as it represents weights used in the original census application not relevant to the task at hand, as well as the `education-num` feature since it is a different representation of the existing `education` feature. This leads to 12 features, namely: `age`, `workclass`, `education`, `marital-status`, `occupation`, `relationship`, `race`, `sex`, `capital-gain`, `capital-loss`, `hours-per-week`, `native-country`. We drop all rows with missing values. When encoding the `native-country` feature we group into "other" all countries other than US and Mexico. We obtain a validation set by randomly splitting off 20% of the remaining training data, leading to a final split of 53.3% training, 33.3 % testing and 13.3% validation data from a total of 45,222 rows. We one-hot encode all categorical features, and standardise the continuous features by subtracting the train set mean and scaling by train set standard deviation.

**COMPAS**

We follow the pre-processing steps used in (Larson et al., 2016) applied to the original dataset[3] which contains 7214 rows and 52 features: We remove rows with charge dates not within 30 days from the arrest date to ensure we relate to the right offence, and only select individuals of African-American and Caucasian race. Some features contain information contained in others, or are not relevant such as defendants' names. We hence select only a small subset of semantically most relevant features, namely: `age`, `sex`, `race`, `c_charge_degree`, `priors_count`, `c_jail_in`,

---

[3]https://github.com/propublica/compas-analysis/

`jail_time_out`, `juv_fel_count`, `juv_misd_count`, `juv_other_count`. The final dataset contains 5278 rows and 10 features, which we randomly split into 60% training, 20% validation and 20% testing data. The categorical variables are one-hot encoded, while the remaining features are standardised by subtracting the train set mean and scaling by train set standard deviation.

## EXPERIMENTS

We here give the details of experimental setup for each of the methods we consider in our experiments in Sec. 3. As much as possible when comparing different methods we attempted to use similar model structures.

### Explainability

To produce the results in Sec. 3.1 we trained a feedforward neural network (one hidden layer, 50 hidden units, and ReLU activations) on the Adult dataset. We then introduced a second neural network with the same architecture to serve as the perturbation. The original model is frozen and we apply the algorithm of Zhang et al. (2018) to train the permutation. We then calculate Shapley values for each of the original model, the perturbation and the corrected model by first sampling coalitions of features, then approximating Eq. (2) – resp. Eq. (6) – via Monte Carlo approximation with the empirical data distribution.

### Robustness

To produce the results of Sec. 3.2 we start with the baseline model of the previous section. We then retrain it to suppress the importance of `sex`. We are motivated by, but deviate slightly from, the work of Dimanov et al. (2020). Whereas they propose adding a penalty term corresponding to the gradient of the loss with respect to `sex`, we instead use a finite difference of the model output with respect to `sex` due to the discrete nature of the feature. Thus the modified loss becomes

$$\frac{1}{N} \sum_{i=1}^{N} \mathcal{L}(f(x_i), y_i) + \alpha \Big| f(x_i \mid \mathrm{do}(\text{sex} = 1)) - f(x_i \mid \mathrm{do}(\text{sex} = 0)) \Big| \tag{20}$$

where $\mathcal{L}$ is the original cross-entropy loss, $\alpha$ is a hyperparameter controlling the trade-off between optimising the accuracy and minimising the effect of `sex`, and $f(x_i \mid \mathrm{do}(\text{sex} = j))$ denotes, via a slight abuse of notation, $f$ evaluated on the data point $x_i$ with the value for `sex` replaced with $j$. We train the baseline model for an additional 200 batches with $\alpha = 3$. The resulting model agrees with the baseline on over 98.5% of the data, and has the same test set accuracy. We calculate Shapley values for the retrained model as before.

### Performance

For Sec. 3.3 we trained a single-layer feedforward neural network on each dataset to serve as a baseline. On the Adult dataset we use 50 hidden units, on COMPAS we used 32. In both cases we gave the network ReLU activations.

We use the training-time algorithms of Agarwal et al. (2018) and Zhang et al. (2018) to train additional neural networks with the same architecture while imposing demographic parity and equalised odds. We used the implementation of the reductions approach of Agarwal et al. from the `fairlearn` library of Bird et al. (2020), with some superficial modifications to make it compatible with our TensorFlow models. We implemented the adversarial approach of Zhang et al. (2018) ourselves. Our implementation runs for a fixed number of iterations, then restores the weights corresponding to the best validation set loss during the second half of training. All models are then evaluated on the unseen test set.

We reapplied both of these algorithms to the problem of learning a perturbation to the baseline model. Again the perturbations were single layer neural networks with the same architecture as the baseline.

Finally we consider two post-processing methods, those of Feldman et al. (2015) [4] and Hardt et al. (2016), to impose demographic parity and equalised odds respectively. We use our own implementation of the former, and the AI Fairness 360 library to apply the latter (Bellamy et al., 2018), each applied to the aforementioned baseline.

---

[4]In the original paper, Feldman et al. (2015) present their algorithm as a pre-processing method. The idea of instead applying it as a post-processing algorithm is not ours, having been previously observed by Hardt `https://mrtz.org/nips17/#/41`. We weren't able to find any additional sources for this idea.

Table 3: Values for hyperparameter grid used in our simulations

| | Variables | Dataset | |
|---|---|---|---|
| | | Adult | COMPAS |
| **Model** | Number of hidden layers | $\{2, 3\}$ | $\{0, 1, 2\}$ |
| | Width of hidden layers | 32 | $\{16, 24\}$ |
| | Learning rate | $\{0.0001, 0.001\}$ | $\{0.0001, 0.001\}$ |
| | Iterations | $\{2500, 5000\}$ | $\{2500, 5000\}$ |
| **Adversary** | Number of hidden layers | $\{2, 3\}$ | $\{0, 1, 2\}$ |
| | Width of hidden layers | 32 | $\{16, 24\}$ |
| | Learning rate | 0.01 | 0.01 |
| | Iterations per model iteration | $\{1, 5\}$ | $\{1, 5\}$ |
| | Batch size | 512 | 128 |

Table 4: Number of observations associated with intervals of increasing fairness

| | | Method | Number of observations in unfairness interval $I =$ | | | |
|---|---|---|---|---|---|---|
| | | | [0.05, 0.1) | [0.025, 0.05) | [0.01, 0.025) | [0, 0.01) |
| **Demographic parity** | **Adult** | Perturbed | 423 | 201 | 146 | 103 |
| | | Zhang et al. | 550 | 281 | 168 | 183 |
| | | Adel et al. | 606 | 214 | 117 | 39 |
| | **COMPAS** | Perturbed | 4009 | 2548 | 1956 | 1459 |
| | | Zhang et al. | 2776 | 2771 | 2526 | 2892 |
| | | Adel et al. | 10015 | 3157 | 1482 | 945 |
| **Equalised odds** | **Adult** | Perturbed | 509 | 1083 | 231 | 76 |
| | | Zhang et al. | 384 | 1140 | 278 | 112 |
| | | Adel et al. | 1554 | 1902 | 334 | 14 |
| | **COMPAS** | Perturbed | 3125 | 2761 | 2854 | 3228 |
| | | Zhang et al. | 1986 | 2381 | 3437 | 4087 |
| | | Adel et al. | 6421 | 6827 | 6330 | 3756 |

**Stability**

To demonstrate instability during training of the adversarial methods, we compare the approaches of Adel et al. (2019), and Zhang et al. (2018), in the latter case applied both as specified by the authors, and also to a perturbation as specified in Sec. 2. Table 3 show the selection of hyperparameters over which we search. We chose them to cover a wide range of plausible experimental configurations. For each combination we additionally vary the weight of the discriminator term in the loss – controlling the relative importance of the fairness objective compared to the accuracy objective. We run 5 experiments with each combination and record the results after a fixed number of iterations to construct the violin plots in Sec. 3.

A.4   STABILITY RESULTS FOR EQUALISED ODDS

We also ran a grid search over hyperparameters with repeats to test the stability of the adversarial training of perturbations as compared to regular adversarial approaches. Fig. 4 shows the distribution of accuracy outcomes for each method on the two datasets we have been considering throughout, Adult and COMPAS. We see similar results to the demographic parity experiments in that the optimal accuracy in each fairness bin is comparable, but there is a much smaller spread of outcomes for the perturbation-based approach. This is because the perturbation remains anchored to a local accuracy optimum throughout, discouraging the model from jumping between local minima. Table 4 shows the number of experimental results that fell in each bin for both the demographic parity, and equalised odds experiments.

A.5   EXPLAINING MODELS THAT DON'T HAVE ACCESS TO PROTECTED ATTRIBUTES

In section Sec. 3.1 we showed model explanations for an unfair model corrected with an additive perturbation trained using the adversarial method of Zhang et al. (2018). In that example, both the

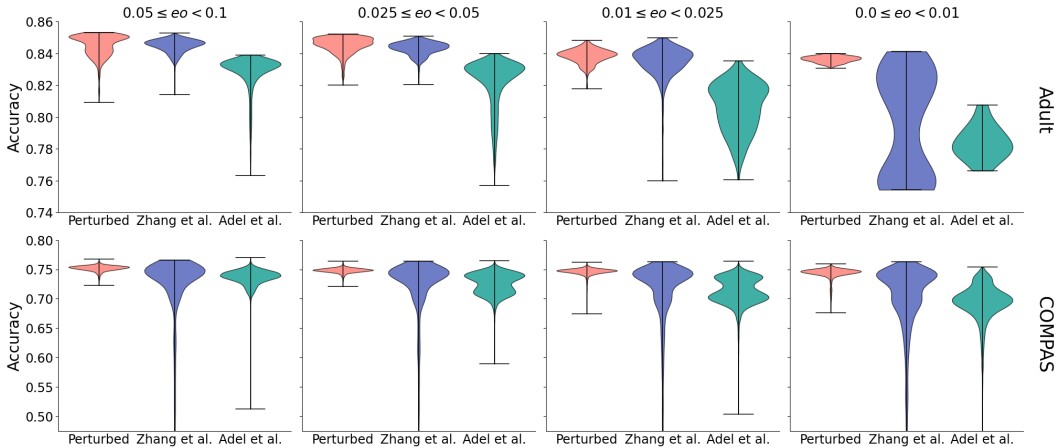

Figure 4: Accuracy violin plots of experimental outcomes binned by achieved level of fairness.

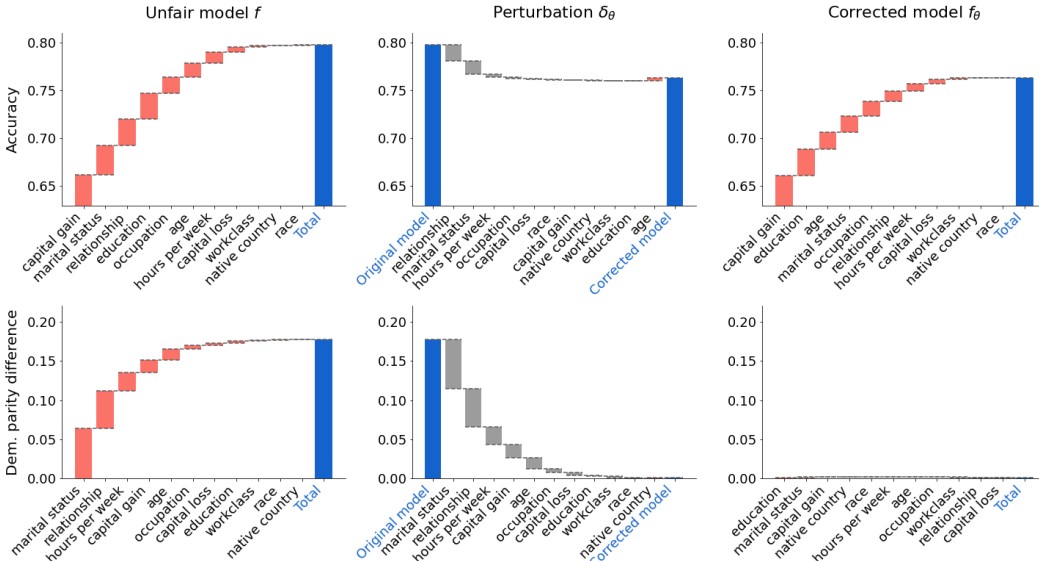

Figure 5: Accuracy violin plots of experimental outcomes binned by achieved level of fairness.

original-unfair model and perturbation had access to the protected attribute. However, this is not a requirement of the adversarial mitigation strategy, and indeed in some real-world scenarios we may not have access to the protected attribute at inference time. In this section we repeat the experiment with an unfair-model and perturbation which do not take the protected attribute as input, and compare the resulting explanations. The experimental setup is unchanged except for the fact that the models no longer take `sex` as input.

The results show some interesting patterns. First we note that both models had comparable accuracy both before and after correction. This is not so surprising as the Shapley values in Fig. 1 show that `sex` is relatively unimportant to accuracy, and it is highly correlated to other features in any case. The baseline model trained without access to the protected attribute was slightly fairer, and the perturbation was slightly less effective in imposting fairness. This is also not so surprising. Witholding the protected attribute from the baseline model should not make it more unfair, but also we expect the perturbation to benefit from access to the protected attribute so that it can make active corrections.

Looking at Fig. 5 it is notable that the fairness Shapley values for the corrected model are all zero. The same Shapley values for the corrected model which had access to the protected attribute were non-zero, but in perfect balance so that their sum was zero. This is because the model that has access to the protected attribute is able to take "affirmative action", which is to say it can actively use the

protected attribute to cancel out the effects of other features. In contrast, the model which does not have access to the protected attribute has no such option, and instead learns to only use information contained in each feature that is "orthogonal" to the protected attribute. As a result no individual feature increases or decreases the disparity between protected groups.

