# OpenReview forum: "Explainability for fair machine learning"
_ICLR.cc/2021/Conference — Reject_

### Official Review · AnonReviewer2 · 2020-10-19

**Rating:** 5
**Confidence:** 4

**Review:**

This paper presents a method for feature attribution for fairness of the classifier.  They also demonstrate a feature augmentation technique to mitigate unfairness.  They connect their attribution method to to the augmentation technique and demonstrate that their method can attribute the necessary changes to achieve fairness. They evaluate their approach on a few tabular data sets.

Introduction + Methods

- I’m slightly confused about equation (2), is the correct interpretation that y is the label we’re explaining? So, say, if we’re explaining the -1 class and f(x) is 0.25, then f_y(x) yields 0.75?
- I think some clarity on the notation would be useful in general.  Is y the ground truth label from the data? That seems to be described this way in section 1.  However, in equation (2) is seems to refer to the label we’re selecting to explain, which feels slightly different.  I think some of this notation could be cleared up a bit.
- “Splicing disjoint sets of features” is this common terminology? I think what this is trying to say is that this step is where we substitute values from the data into a point to create a perturbation.  Let me know if I am correct.
- For the description for equation (5): “…the expected accuracy for a model which samples a predicted label according to the predicted probability” what does this mean? The notation in equation (5) looks like it’s the expected prediction of the model, where is the relation to accuracy here? I’m not quite seeing this.
-  I think that equation (5) could be clarified in general.  It would be good to explicitly state this significance of this property.  (It’s stated clearly for equation 9, and stating it explicitly here would help readers)
- Is g a value function? If so, it’d be useful to state this leading up to the equation. Also, we’ve introduced the term “a” at this point which refers to the protected attribute (going back to the introduction).  What values can a take on? Are we assuming $a \in \{ 0, 1\}? It would be good to clarify this. Last what is p(a) meant to refer to? Is this the overall proportion of individuals with a certain protected attribute? If so, why do we need this term?
- “The linearity axiom of the Shapley values guarantees that the fairness Shapley values of a linear ensemble of models are the corresponding linear combination of Shapley values of the underlying models” — it would be good to clarify this sentence as it’s slightly confusing right now.
- For equation (10) assuming that f Is a linear model, is $\delta_\theta$ meant to be a vector of values added to each of it’s coefficients?
- Is is slightly contradictory that in equation (10) $\delta_\theta$ is a perturbation (and can be explicitly added to f) but then becomes a function in equation (11)? I get what is being said but perhaps the notation could be improved here.

Experiments
- Figure 1 is hard to read, can you make the labels bigger for each graph. Also, it might be good to place the y axis values on each graph so that its easier to see the scale.  I see what the graph is saying, but this would make it much easier to figure out.
- I think figure 1 is a nice visualization overall and gets the point across well.
- I’m not fully understanding the point being made by section 3.2.. Is the idea that we know this suppressed model doesn’t rely on sex at all, so the fairness Shapley values shouldn’t say it does?

General Questions + Comments:
- I liked the ideas presented by this paper overall.  The idea of using Shapley values to provide feature attributions for fairness is interesting — particularly the applications for explaining the differences between two models, one of which has been corrected for fairness.
- I do feel however that the paper can be improved in a number of places in order to strengthen the work as mentioned in my comments.  It would be very useful if the authors provided answers to some of my questions and took some of the comments into consideration for a revision. Particularly, I'm somewhat confused about the contribution of section 3.2 and would appreciate clarification.
- One related question is that the authors demonstrate their method on the suppression techniques from Dimanov et al.  Would their techniques help at all with the related methods (more so phrased as attacks) from https://arxiv.org/abs/1911.02508?
- Overall, my sentiments right now are borderline, leading towards reject. There’s a number of points which could warrant further clarification right now in the paper.

---

> ### Author Response · Authors · 2020-11-16
> **Response to R2 - Clarity improvements and answers to questions**
>
> We thank the reviewer for a careful reading of our paper and the many helpful comments and questions.
>
> As discussed in our responses to the other reviewers and the general comment, we have made a number of edits to the paper to improve clarity in key sections, including the addition of a section with setup and notation. We hope this addresses a number of the points of confusion.
>
> We now address specific points made by the reviewer. Any points we do not address directly should be assumed to be remedied by the clarity improvements.
>
> **Introduction + Methods**
> * Your interpretation is correct, we have made this explicit in the text
> * We have also clarified this. The goal is to explain the predicted probability of the ground truth label (so for different data points we might explain a different component of the output of $f$). Therefore in both cases $y$ was being used to mean the same thing.
> * “Splicing” is commonly used terminology in this context, but we have reworded the description to be more explicit.
> * We have also reworded this description. The first term in equation (5) (now (4)) is the average probability the model assigns to the correct label. This is therefore equivalent to the proportion of correct predictions you would expect if you sampled labels from the predicted probabilities. In other words it represents the accuracy of a randomised classifier.
> * We have clarified the notation and role of a in the definition of $g$. The factor of $p(a)$ is needed to counteract the effect of different protected group sizes when aggregating to get global Shapley values.
> * Delta is a function, and equation (10) (now (9)) refers to addition of functions, rather than modification of the internal weights / parameters of the model $f$. We have updated the notation to make this clearer. This also answers the reviewers next question about $\delta$ being written as a function in the following equation.
>
> **Experiments**
> * We have made the labels larger and added tick labels to each axis as suggested.
> * In section 3.2 we seek to show that fairness Shapley values cannot be manipulated to hide unfairness in the same way that regular Shapley values can [1]. The key observation is that while it is still possible to manipulate individual Shapley values, collectively the Shapley values are constrained to sum to the demographic parity difference, and hence the only way to hide unfairness from the fairness Shapley values is to eliminate it. We additionally observe that the attribution of unfairness to individual features allows us to understand what is happening in the manipulated model: namely the model has shifted focus instead to close proxies of sex, and as a result should not be considered to be fair. We have clarified some of the confusing exposition in this section.
>
> **General Question + Comments:**
> * Adversarial fooling of explanation methods. The effect of the adversarial attacks described in [2] would likely be similar to our findings with the suppression technique of [1]. As noted above, the Shapley values must sum to the unfairness, so at best any attack could manipulate individual Shapley values, but would not be able to give the illusion of a fair model without actually making the model fair. As a side note, the attack described in [2] exploits the fact that Shapley explanations rely on model evaluations on data that lies off the data manifold / out of distribution. The fix to this is to use an on-manifold approximation of the value function as described in [3] and mentioned in footnote 1 of our paper. This is not the focus of our paper, and the changes required are simple so we did not include the details in our write-up.
>
> We hope these answers clear up any outstanding questions, and believe that the revisions we have made address the stated clarity concerns.
>
> [1]: Dimanov, Botty, et al. "You Shouldn't Trust Me: Learning Models Which Conceal Unfairness From Multiple Explanation Methods." SafeAI@ AAAI. 2020.
>
> [2]: Slack, Dylan, et al. "Fooling lime and shap: Adversarial attacks on post hoc explanation methods." Proceedings of the AAAI/ACM Conference on AI, Ethics, and Society. 2020.
>
> [3]: Frye, Christopher, et al. "Shapley-based explainability on the data manifold." arXiv preprint arXiv:2006.01272 (2020).

---

### Official Review · AnonReviewer4 · 2020-10-29
**A good contribution connecting the fields of explainability and fairness.**

**Rating:** 6
**Confidence:** 3

**Review:**

Quality
- This paper has defined a well-scoped problem: explaining the unfairness of an ML model in terms of the features used, as well as explaining an additive perturbation that will make the model more “fair”.
- This paper acknowledges that definitions of fairness can be fraught and should be applied with care and domain knowledge. Therefore it has proposed an “explanation” procedure that works with multiple statistical definitions of fairness. This is a strength of the work.


Clarity
- Clarity of the “section 2” could be improved. What is the support of $a$, $y$ and $f(x)$?

Originality
- There has not been a lot of work on fairness and explainability. As one of the first forays into these questions showing a positive result (as claimed in introduction, and also to my best knowledge), I think the paper is sufficiently original. Even though it is adapting a well-known construct (Shapley value), its originality also lies in connecting explanations to work on post-hoc fairness corrections (section 2.2).

Significance
- As stated above, I think the paper makes a significant contribution to research on fairness and explainability by developing some basic tools to help with “explaining” group fairness issues with ML models.
- That being said, I think one must be careful to claim that the explanations offered by shapley value suggest any particular intervention. For example, it is not true that the features that contribute most to unfairness should be always be removed. I think a careful discussion on how to use the insights from the shapley values in practice, or some open questions regarding the interpretation of the shapley values would improve this paper.

---

> ### Author Response · Authors · 2020-11-16
> **Response to R4 - Clarity improvements and discussion of use case**
>
> We thank the reviewer for a thoughtful summary of our work and positive comments.
>
> The reviewer cited clarity of section 2 as a concern. As noted in our general response and response to the other reviewers we have made a number of edits to the paper to address some of the issues. In particular we have made explicit the supports of $a$, $y$ and $f(x)$ as requested.
>
> We appreciate the comment that we should be careful to claim that the explanations presented in our work do not suggest any particular intervention. This is very much aligned with our view, and in fact we are sceptical that selection of a fairness metric could or even should be automated. Instead we view selection of a fairness metric or intervention to be a choice that is heavily dependent on a good understanding of the context. We believe that fairness-specific explanations can help with understanding the context, and so are a valuable component in choosing the right notion of fairness and an intervention. By themselves we would not consider them sufficient to make such decisions responsibly. It is worth considering the counterfactual, without these explanations interventions have to be selected on the basis of high-level metrics only. Attributing these metrics to individual features is relevant information that can help with downstream decision making.
>
> We believe the revisions fully address the reviewer's primary concern of a lack of clarity, and hope that they agree.

---

### Official Review · AnonReviewer1 · 2020-10-29
**Important topic, but lacks calrity**

**Rating:** 5
**Confidence:** 4

**Review:**

The goal of the paper is to design mechanisms to explain the unfairness in the outcomes of a ML model and propose methods to mitigate unfairness. The paper uses the Shapley value framework. The main idea is to alter the prediction function so that instead of providing the classification score, an "unfairness" score is returned. An out of the box application of the Shapley value framework on this unfairness score now returns the "unfairness" feature attribution. These feature attributions can be used to explain the unfairness of the model. The paper then proposes to learn a linear perturbation, which when combined with the additive property of Shapley framework results in updated "unfairness" attributions.

While the paper tackles an interesting and timely problem, it lacks clarity at several important points. Additionally, some of the claims seem to be a little overreaching. The experimental evaluation can also use some more thoughtful analysis. Please see detailed comments and suggestions for improvement below:

1- First of all, I would highly recommend setting aside a separate (sub)section to describe the setup. Right now, the details about whether f(x) is a real number of a probability, whether y is {0,1} or {-1,1}, or what "a" is are scattered across the text, reducing the readability of the paper.

2- Going from Eq. 2 to Eq. 5, the paper somehow switches from prediction probabilities to accuracies. Shouldn't there be a threshold function to convert the probabilities into predictions? Or is the paper only focusing on randomized classifiers? If only the randomized classifiers are the focus, if/how does it limit the extensibility of the proposed technique?

3- Does it make any difference for the explanations when the probabilities are high non-calibrated (https://arxiv.org/pdf/1706.04599.pdf)?

4- What was the accuracy/generalizability of this $\delta$ in the experiments?

5- After reading Section 2.2, a reader would think: Why even train the linear perturbation? Why not simply train a separate fair model (from the same model class) and get the Shapley predictions of that second model? I think the main idea here is that the paper tries to leverage the Shapley additive property here. How precisely does the additive property help?

6- The results in Figure 1 are interesting, but not very surprising. Specifically, given that one achieves roughly the same accuracy on Adult data regardless of whether the sensitive feature is used or not, it would have been more interesting (and insightful) to show the Shapley plots for the cases when the sensitive feature is indeed not used for classification.

7- The claim made in the paper that fairness is achieved at "no loss of accuracy" is perhaps a bit too bold given that there are well-known results about fairness-accuracy tradeoffs (e.g., https://arxiv.org/abs/1701.08230 and https://arxiv.org/abs/1609.05807).

--------------------
Post rebuttal comments:

Thanks to the authors for the helpful comments -- they indeed help clarify some of the confusions. As a result, I have upgraded my score. However, I am still leaning towards reject because I feel there are still open questions that may hinder the adaptability of the proposed method in the real world. Specifically, given the response to question 3 above, it would help to know what are the real world situations where one uses a randomized classifier and is still interested in model interpretability (the two seem to be at odds with each other as randomness inherently seems a bit arbitrary). Another concern that I have is about Eq. 3 in the paper: Why is the sum function chosen to compute global explanations from local ones? There seem to be multiple ways to do this (e.g., median, sum of absolute values) and it would help to know what are the (dis)advantages of not using other aggregation functions.

---

> ### Author Response · Authors · 2020-11-16
> **Response to R1 - Clarity improvements and explanation of claim**
>
> We appreciate the reviewer’s careful summary of our paper and thoughtful comments.
>
> The reviewer’s primary concern was the lack of clarity. This was shared with the other reviewers, so we have made a number of edits to the paper in order to address these concerns. See the general comment for more details. We will focus here on addressing the reviewer’s specific comments. We would particularly like to bring attention to the reviewer's point 7. Here there was concern we were overreaching with our claims, in fact the claim we were making is weaker but was not stated clearly. We have amended the text and explained the true claim below.
>
> 1 - We thank the reviewer for this suggestion and have added an explicit subsection for notation and setup.
>
> 2 - We have made it clearer in the text that we are referring to the accuracy of a randomised classifier. This does not limit the extensibility of our work to non-randomised classifiers in our view. Firstly, the randomised accuracy of a non-randomised classifier is still a valid quantity of interest, blending accuracy and model confidence in a single metric. Secondly the value function can be adjusted to use model predictions instead of model predicted probabilities. In this case the Shapley values sum to accuracy and demographic parity respectively. These value functions are non-linear, which compromises the linearity axiom, hence we opted for the value functions described in the paper.
>
> 3 - The Shapley values sum to randomised accuracy which depends both on accuracy and the model confidence. Accuracy and confidence of course interact via calibration. Consider two 100% accurate models, one which predicts with 100% confidence (and hence is perfectly calibrated) and another which predicts with 51% confidence always. Though these models have the same accuracy, the latter will have a randomised accuracy of only 51%. So calibration will have an effect, but it is better to think of randomised accuracy via accuracy and confidence.
>
> 4 - Tables 1 and 2 show the accuracy of perturbed models on two test datasets, with comparisons to several baselines. The tables show that the perturbed models generalised well to unseen data.
>
> 5 - Leveraging the additive property is indeed the point. It allows us to construct three related models on the data with consistent explanations, that together help us understand unfairness from multiple perspectives: sources of unfairness for a model trained without intervention, the change that was required to make the model fair, and fairness within the corrected model.
>
> 6 - This is an interesting question. We did experiment with models trained without access to the protected attribute. As one might expect, the original model is a little fairer, and the correction is a little less effective. Ultimately we decided to use the current experiment in the main body of the paper. However since the same experiment without the protected attribute is of interest we have included a figure and discussion in the supplementary material.
>
> 7 - This is not a claim we intended to make. When we say there is “no loss in accuracy” we mean that applying training-time fairness algorithms to an additive correction performs as well as applying the same training-time fairness algorithm to a new, unconstrained model. We have clarified this in the text to avoid future confusion.
>
> We believe the submitted improvements address the primary concern of a lack of clarity, and hope that the reviewer agrees.

---

### Author Response · Authors · 2020-11-16
**General response - clarity improvements**

We thank all of our reviewers for careful consideration of our paper. A common theme to all reviews was a lack of clarity in certain key parts of the paper. We have made a number of improvements, most notably the addition of a background / notation subsection at the start of section 2. We additionally have clarified assumptions or wording in a number of places in the text, and improved the readability of one of the main figures. Though the content is ultimately unchanged, we believe the clarity is far improved. We hope the reviewers agree, and thank them for specific examples pointed out in their reviews, all of which we have addressed.

---

### Decision · Program_Chairs · 2021-01-07
**Final Decision**

**Decision:**

Reject

**Comment:**

All the reviewers found interesting the use of Shapley values to provide feature attributions for fairness, however, the reviewers brought up a number of issues, particularly in terms of presentation and clarity. While the authors' responses did clarify some of these concerns, this was not enough for the reviewers to broadly support acceptance.